# Single Nucleotide Polymorphisms in the Vitamin D Metabolic Pathway and Their Relationship with High Blood Pressure Risk

**DOI:** 10.3390/ijms24065974

**Published:** 2023-03-22

**Authors:** Susana Rojo-Tolosa, Noelia Márquez-Pete, José María Gálvez-Navas, Laura Elena Pineda-Lancheros, Andrea Fernández-Alonso, Cristina Membrive-Jiménez, María Carmen Ramírez-Tortosa, Cristina Pérez-Ramírez, Alberto Jiménez-Morales

**Affiliations:** 1Pharmacogenetics Unit, Pharmacy Service, University Hospital Virgen de las Nieves, Avda. de las Fuerzas Armadas 2, 18004 Granada, Spain; 2Department of Biochemistry and Molecular Biology II, Faculty of Pharmacy, Campus Universitario de Cartuja, University of Granada, 18011 Granada, Spain; 3Biosanitary Research Institute ibs.GRANADA, Avda. de Madrid, 15, 18012 Granada, Spain; 4Cancer Registry of Granada, Andalusian School of Public Health, Carretera del Observatorio, 4, 18011 Granada, Spain

**Keywords:** high blood pressure, vitamin D, risk, genetic polymorphisms, *CYP2R1*, *CYP27B1*, *CYP24A1*, *GC*, *VDR*

## Abstract

High blood pressure (HBP) is the leading risk factor for cardiovascular disease (CVD) and all-cause mortality worldwide. The progression of the disease leads to structural and/or functional alterations in various organs and increases cardiovascular risk. Currently, there are significant deficiencies in its diagnosis, treatment, and control. Vitamin D is characterized by its functional versatility and its involvement in countless physiological processes. This has led to the association of vitamin D with many chronic diseases, including HBP and CVD, due to its involvement in the regulation of the renin–angiotensin–aldosterone system. The aim of this study was to evaluate the effect of 13 single nucleotide polymorphisms (SNPs) related to the vitamin D metabolic pathway on the risk of developing HBP. An observational case-control study was performed, including 250 patients diagnosed with HBP and 500 controls from the south of Spain (Caucasians). Genetic polymorphisms in *CYP27B1* (rs4646536, rs3782130, rs703842, and rs10877012), *CYP2R1* rs10741657, *GC* rs7041, *CYP24A1* (rs6068816, and rs4809957), and *VDR* (BsmI, Cdx2, FokI, ApaI, and TaqI) were analyzed by real-time PCR using TaqMan probes. Logistic regression analysis, adjusted for body mass index (BMI), dyslipidemia, and diabetes, showed that in the genotypic model, carriers of the *GC* rs7041 TT genotype were associated with a lower risk of developing HBP than the GG genotype (odds ratio (OR) = 0.44, 95% confidence interval (CI): 0.41–0.77, *p* = 0.005, TT vs. GG). In the dominant model, this association was maintained; carriers of the T allele showed a lower risk of developing HBP than carriers of the GG genotype (OR = 0.69, 95% CI: 0.47–1.03; TT + TG vs. GG, *p* = 0.010). Finally, in the additive model, consistent with previous models, the T allele was associated with a lower risk of developing HBP than the G allele (OR = 0.65, 95% CI: 0.40–0.87, *p* = 0.003, T vs. G). Haplotype analysis revealed that GACATG haplotypes for SNPs rs1544410, rs7975232, rs731236, rs4646536, rs703842, and rs10877012 were associated with a marginally significant lower risk of developing HBP (OR = 0.35, 95% CI: 0.12–1.02, *p* = 0.054). Several studies suggest that *GC* 7041 is associated with a lower active isoform of the vitamin D binding protein. In conclusion, the rs7041 polymorphism located in the *GC* gene was significantly associated with a lower risk of developing HBP. This polymorphism could therefore act as a substantial predictive biomarker of the disease.

## 1. Introduction

The European Society of Cardiology and the European Society of Hypertension define arterial hypertension or high blood pressure (HBP) in adults as systolic blood pressure (SBP) greater than or equal to 140 mmHg or diastolic blood pressure (DBP) greater than or equal to 90 mmHg, measured in the consulting room [1,2]. The American College of Cardiology/American Heart Association, for their part, locate the reference values as SBP/DBP greater than or equal to 130/80 mmHg [3]. The societies mentioned, together with the World Health Organization (WHO), agree that HBP is the main risk factor for developing cardiovascular disease (CVD) and for premature mortality [1,2,3,4,5,6]. According to the data reported by the non-communicable disease risk factor collaboration (NCD-RisC) and the WHO for the year 2019, the prevalence of HBP in adults was 32% in women and 34% in men, and it is estimated that there are 1278 million people with the disease [4,6]. 

There is a wide range of modifiable risk factors that contribute to the development of HBP, including age, sex, an unbalanced diet, overweight, obesity, alcohol abuse and smoking, vitamin D deficiency, lack of physical activity, psychological stress, socioeconomic factors, and inadequate access to healthcare [2,3,4,5,6]. Given the diversity of symptoms and complex etiology of HBP, in most patients with this disease, no underlying cause is detected [2,3,4,5,6].

The existence of a causal relationship between vitamin D and HBP is due to the participation of the active isoform of vitamin D (calcitriol, 1,25-dihydroxyvitamin D or 1,25-(OH)_2_D) in regulating the renin–angiotensin–aldosterone system (RAAS) by inhibiting expression of the gene that codes for renin, a key component in controlling blood pressure (BP) [6,7,8,9]. Moreover, given the ability of vitamin D to regulate the immune system and its anti-inflammatory activity, a deficiency of this vitamin is associated with the triggering of cytokine-mediated inflammatory processes, which lead to endothelial dysfunction and increased stiffness, contributing to raised BP [9,10,11,12,13,14,15,16].

The activity of vitamin D depends on its complex metabolism being carried out correctly. The term vitamin D comprises both the active and the inactive isoforms. The inactive ones are cholecalciferol and ergocalciferol. The first is synthesized in the dermis through the action of ultraviolet (UV) radiation, especially UVB (290–320 nm), on 7-dehydrocholesterol, while both the first and the second are incorporated through the diet [9,17,18]. Both molecules are transported in the blood bound to vitamin D binding protein (VDBP) or GC, which carries them to the liver, where the CYP2R1 or 25-hydroxylase enzyme carries out a hydroxylation at position 25 of the molecule, producing calcidiol or 25(OH)D. This is the metabolite that remains in the blood for longer and is, therefore, the one that is measured when determining serum levels [19,20,21]. It then travels to the kidney, where the CYP27B1 or α-1-hydroxylase enzyme includes a hydroxyl group at position 1, giving rise to calcitriol. To perform the vitamin’s activity, calcitriol binds to the vitamin D receptor (VDR), found in the cell membrane [19,20,21,22]. Once they are bound, the VDR translocates to the nucleus to form a complex with the retinoid X receptor (RXR), an orphan retinoid/steroid hormone receptor, which will act as a transcription factor in the expression of various genes involved in numerous physiological phenomena [9,17,18,19,20,21,22,23]. Finally, calcitriol will be degraded by successive hydroxylation reactions catalyzed by the CYP24A1 enzyme, located in the mitochondria, with the object of increasing the solubility of the molecule and eliminating it by renal excretion [20,21,22,23]. 

The genes which code for the transporter, enzymes, and receptor that participate in vitamin D metabolism are characterized as being highly polymorphic. Consequently, the presence of single nucleotide polymorphisms (SNPs) in the *GC*, *CYP2R1*, *CYP27B1*, *CYP24A1*, and *VDR* genes influences serum levels of vitamin D, and hence its activity [9,23,24,25,26,27,28]. Therefore, the presence of SNPs in the genes mentioned may influence the development of HBP. 

This study was designed with the aim of evaluating the SNP-type genetic polymorphisms in the genes involved in vitamin D metabolism and their relationship to the risk of suffering from HBP in Caucasian patients, specifically those residing in southern Spain.

## 2. Results

### 2.1. Study Subjects Characteristics

A total of 750 study participants were included: 250 cases diagnosed with HBP and 500 controls. Their sociodemographic and clinicopathological characteristics are described in Table 1.

The group of cases comprised 52.4% (131/250) women, and the median age was 66 (range: 60–73) years. With regard to smoking status, 60.7% (147/242) were non-smokers, 33.1% (80/242) were ex-smokers, and 25.2% (61/242) current smokers. In terms of drinking status, 71.2% (163/229) were classified as non-drinkers, 26.6% (61/229) as current drinkers, and 2.2% (5/229) as ex-drinkers. Most of the HBP patients were categorized as obese (44.5%, 94/211), 39.3% as overweight, and 16.1% as normal weight. A total of 50.8% (127/250) did not have dyslipidemia, and 58.4% (146/250) did not suffer from diabetes. As for the clinicopathological features of the disease, 58.4% (146/250) showed elevated SBP levels, and 58.4% (146/250) had normal DBP levels; total cholesterol and LDL cholesterol levels were normal in 55% (137/249) and 63.7% (142/223), respectively, and HDL cholesterol levels showed medium values in 53.3% (121/227); 69.2% (171/247) had normal triglyceride levels; mean glucose was 114 ± 47 mg/dL.

The control group was made up of 51.4% (257/500) women, and the median age was 65 (range: 60–73) years. There were 51.9% (246/474) non-smokers, 25.5% (121/474) ex-smokers, and 22.6% (107/474) current smokers. With regard to alcohol drinking status, 77.7% (351/452) were classified as non-drinkers, 18.4% (83/452) as current drinkers, and 4% (18/452) as ex-drinkers. The majority (39.8%) were overweight (135/339), 32.7% (111/339) were normal weight, and 27.4% (93/339) were obese. A total of 73.4% (367/500) had no dyslipidemia, and 89.1% (434/487) did not suffer from diabetes.

There were significant differences between the cases and the controls with respect to drinking status (odds ratio (OR) = 1.58, 95% confidence interval (CI): 1.08–2.31, *p* = 0.027, for drinker vs. non-drinker; and OR = 0.60, 95% CI: 0.19–1.53, *p* = 0.027, for ex-drinker vs. non-drinker), body mass index (BMI) (OR = 3.30, 95% CI: 2.06–5.38, *p* < 0.001, for obesity vs. normal weight and OR = 2.01, 95% CI: 1.26–3.24, *p* < 0.001, for overweight vs. normal weight), dyslipidemia (OR = 2.67, 95% CI: 1.95–3.68, *p* < 0.001), and diabetes (OR = 5.83; 95% CI: 4–8.58, *p* < 0.001), respectively. No statistically significant differences were observed for sex (*p* = 0.796) or age (*p* = 0.989).

### 2.2. Genotype Distributions

The observed genotype frequencies did not significantly deviate from the expected frequency values according to the Hardy–Weinberg Equilibrium test. With the exceptions of *CYP27B1* rs3782130 (*p* = 0.002769), *CYP27B1* rs703842 (*p* = 0.03971), and *CYP24A1* rs6068816 (*p* = 0.03246) for the control group (Appendix A). For these three variants, we found no statistically significant differences from those described in the Iberian population: *CYP27B1* rs3782130 C allele: 0.3050 vs. 0.2850, *p* = 0.9753; *CYP27B1* rs703842 C allele: 0.2507 vs. 0.2940; *p* = 0.9452; and *CYP24A1* rs6068816 T allele: 0.1109 vs. 0.1070; *p* = 0.9929, respectively [29]. The linkage disequilibrium (LD) r^2^ and D’ values are shown in Appendix A. The SNP pairs rs4646536-rs10877012 (r^2^ = 0.731762, D’ = 0.905063), rs4646536-rs703842 (r^2^ = 0.747667, D’ = 0.89186), and rs10877012-rs703842 (r^2^ = 0.758471, D’ = 0.904393) were in strong LD based on D’ values (Figure 1). All the polymorphisms showed minor allele frequencies (MAFs) greater than 1%, and none of them were excluded from the analysis (Appendix A). The haplotype frequency estimates are presented in Appendix A.

### 2.3. Influence of Genetic Polymorphisms on the Risk of HBP

The bivariate analysis between the 13 genetic polymorphisms included in the study and the risk of HBP was evaluated through various analytical models: genotypic, allelic, dominant, recessive, and additive (Appendix A). We observed a statistically significant association for the *GC* rs7041 polymorphism in the additive (*p* = 0.028) and allelic (*p* = 0.031) models, and a tendency toward statistical significance in the dominant (*p* = 0.054) and genotypic (*p* = 0.088) models (Appendix A). In the allelic model, the T allele was associated with a lower risk of HBP (OR = 0.79, 95% CI: 0.63–0.98, T vs. G), and the additive model was in line with this association (OR = 0.78, 95% CI: 0.62–0.97). In the genotypic model, carriers of the *GC* rs7041 TT genotype were associated with a lower risk of developing HBP (OR = 0.61, 95% CI: 0.39–0.95, TT vs. GG), as were carriers of the *GC* rs7041 TG genotype (OR = 0.76. 95% CI: 0.53–1.10, TG vs. GG). In addition, in the dominant model, it was observed that those individuals carrying the T allele showed a lower risk of suffering from HBP (OR = 0.72; 95% CI: 0.51–1.00; TT + TG vs. GG) (Table 2). The logistic regression analysis adjusted for BMI, dyslipidemia, and diabetes showed that in the genotypic model, carriers of the *GC* rs7041 TT and TG genotypes were associated with a lower risk of developing HBP (OR = 0.44, 95% CI: 0.41–0.77, *p* = 0.005, TT vs. GG; and OR = 0.64, 95% CI: 0.41–0.99, *p* = 0.045, TG vs. GG). This association was maintained in the dominant model, where carriers of the T allele showed a lower risk of developing HBP (OR = 0.69, 95% CI: 0.47–1.03, *p* = 0.01, TT + TG vs. GG). The recessive model showed that the *GC* rs7041 TT genotype presented a lower risk (OR = 0.58, 95% CI: 0.35–0.95, *p* = 0.035, TT vs. TG + GG). Finally, the additive model, in line with the previous models, revealed the *GC* rs7041 T allele was associated with a lower risk of suffering from HBP compared to the G allele (OR = 0.65, 95% CI: 0.40–0.87, *p* = 0.003, T vs. G) (Table 3). The remaining SNPs studied did not show a statistically significant association with the risk of developing HBP in any of the models analyzed (Appendix A). The haplotype analysis was performed taking into account those polymorphisms that were in strong LD, and it was discovered that the GACATG haplotype—for the 6 SNPs located on Chromosome 12, i.e., rs1544410 (*VDR*), rs7975232 (*VDR*), rs731236 (*VDR*), rs4646536 (*CYP27B1*), rs703842 (*CYP27B1*), and rs10877012 (*CYP27B1*)—was associated with a lower risk of developing HBP (OR = 0.35; 95% CI: 0.12–1.02, *p* = 0.054) (Table 4).

## 3. Discussion

Arterial hypertension, which is also commonly referred to as HBP, is the main avoidable risk factor for CVD and all-cause mortality. Recent studies have related vitamin D deficiency with HBP and CVD and suggested that vitamin D serum status is a predictive factor of cardiovascular morbidity and mortality [30,31,32,33,34,35]. This study was conducted with the objectives of evaluating the impacts of various SNPs involved in the vitamin D metabolic pathway individually on HBP risk and assessing the haplotypic associations of six SNPs located on Chromosome 12 with risk of HBP.

The results obtained in this study showed that the T allele of the *GC* rs7041 SNP was associated with a lower risk of HBP in all the models studied. Several reviews indicate that the T allele of the *GC* rs7041 is associated with the slower transport phenotype of VDBP [36,37]. Up to now, the relationship between *GC* rs7041 and the risk of suffering from HBP has been studied by only very few studies. However, there are studies in which the presence of this polymorphism is related to the development of CVD. The study carried out by Kiani et al. (2019), with 249 cases and 182 controls in a Caucasian population (Iran) diagnosed with heart disease by angiography, indicated that individuals carrying the TG + TT genotypes in *GC* rs7041 SNP in the dominant model had higher SBP/DBP levels than those with the TT genotype (*p* < 0.01) [38]. These results are in line with our study, in which the T allele showed a protective effect against the risk of HBP. On the other hand, the study conducted by Daffara et al. (2017) in a cohort composed of 1080 Caucasian patients (Italy) found no relationship between the prevalence of heart disease and the *GC* rs7041 SNP (OR = 1.26, 95% CI: 0.73–2.19, *p* = 0.41, GT vs. TT; and OR = 1.25, 95% CI: 0.82–1.91, *p* = 0.30, GG vs. TT) [39]. Similarly, Michos et al. (2015), in their study comprising a cohort of 11945 Caucasian and African American patients (United States) participating in the Atherosclerosis Risk in Communities (ARIC) Study, in which the *GC* rs7041 SNP and vitamin D levels were assessed in relation to coronary heart disease (CHD), found no statistically significant interaction between 25(OH)D and the *GC* rs7041 SNP in relation to risk of CHD (*p*-interaction = 0.87) [40]. 

In our study, no statistically significant associations were observed for the five *VDR* gene SNPs examined, i.e., rs731236, rs7975232, rs1544410, rs2228570, and rs11568820, in any of the models analyzed (Appendix A). In line with our results is the meta-analysis by Zhu et al. (2019) in populations of diverse ancestries (China, Italy, United States, and India) [7]. For *VDR* rs2228570 (FokI) SNP, the meta-analysis included 4011 cases and 4847 controls from populations of diverse ancestries (China, Italy, United States, and India) showed no association with greater susceptibility to HBP is found in any of the models ((OR = 1.02, 95% CI: 0.75–1.38, *p* = 0.90, and I^2^ = 86%, *p*(I^2^) < 0.0001, TC + CC vs. TT), (OR = 0.98, 95% CI: 0.88–1.09, *p* = 0.68, I^2^ = 32%, *p*(I^2^) = 0.18, CC vs. TT + TC), (OR = 0.97; 95% CI: 0.77–1.23, *p* = 0.82, I^2^ = 78%; *p*(I^2^) = 0.0001, TC vs. TT + CC), (OR = 1.02, 95% CI: 0.84–1.23, *p* = 0.86, I^2^ = 82%, *p*(I^2^) < 0.0001, T vs. C)). Nor has any association been demonstrated between the *VDR* rs7975232 (ApaI) polymorphism in two Asian populations (both from China), with 517 cases and 355 controls, and the development of HBP in any of the models ((OR = 1.05, 95% CI: 0.84–1.32, *p* = 0.66, I^2^ = 36%, *p*(I^2^) = 0.21, GT vs. TT vs. GG), (OR = 1.21, 95% CI: 0.54–2.71, *p* = 0.64, I^2^ = 70%; *p*(I^2^) = 0.07, TT vs. GG + GT), (OR = 0.95; 95% CI: 0.54–1.67, *p* = 0.87, I^2^ = 81%; (I^2^) = 0.02, GT vs. GG + TT), (OR = 1.04, 95% CI: 0.87–1.23, *p* = 0.68, I^2^ = 0%, *p*(I^2^) = 0.83, G vs. T)). However, in three populations (from United States, China, and Italy), with 826 cases and 627 controls, the presence of the genotypes GA + AA of the *VDR* rs1544410 (BsmI) SNP as a risk factor for HBP was evident in the dominant model, whereas the presence of the heterozygous genotype GA of the *VDR* rs1544410 (BsmI) SNP was a risk factor for HBP in the heterozygous model compared to the homozygous genotypes ((OR = 1.32, 95% CI: 1.05–1.68, *p* = 0.02, I^2^ = 39%, *p*(I^2^) = 0.19, GA + AA vs. GG), (OR = 1.27, 95% CI: 1.01–1.60, *p* = 0.04, I^2^ = 41%, *p*(I^2^) = 0.18, GA vs. GG + AA, respectively)) [7]. By contrast, Muray et al. (2003) conducted a cross-sectional study of 590 apparently healthy subjects of Caucasian origin (Spain), which showed the presence of the GG genotype of the *VDR* rs1544410 (BsmI) SNP as a risk factor for HBP in men (*p* < 0.006) [41]. 

With respect to the SNPs present in the *CYP27B1* gene, the study conducted by Wang et al. (2014), with the use of genome-wide genotype data in a Caucasian population (23,294 cases, Europe) to evaluate the impact of the polymorphisms involved in the vitamin D signaling and metabolic pathways on BP, concluded that the *CYP27B1* rs4646536 and *CYP27B1* rs703842 polymorphisms are not related to HBP (*p* > 0.05) in the Women’s Genome Health Study (WGHS) or International Consortium of Blood Pressure (ICBP), respectively [42]. These results are in line with those shown in our study, where no statistical relationship was found between the presence of those SNPs and the risk of suffering from HBP.

As for the SNPs in the *CYP2R1* gene, no significant relationship between *CYP2R1* rs10741657 and the risk of HBP was found in our study. In contrast to our findings, Ye et al. (2019) described the presence of this SNP as a protective factor against the risk of HBP in an Asian population (n = 324 cases/525 controls, China). This study showed how the presence of the *CYP2R1* rs10741657 TT genotype acted as a protective factor, regardless of vitamin D levels, for the additive model (OR = 0.81, 95% CI: 0.66–0.98, *p* < 0.05) and for the dominant model (OR = 0.73, 95% CI: 0.56–0.97, *p* < 0.05) [43].

Finally, in our study, the SNPs located in the *CYP24A1* gene (rs6068816 and rs4809957) were found not to be associated with the risk of suffering from HBP. The results shown by the previous study conducted by Ye et al. (2019) in an Asian population are in line with our results, since none of the polymorphisms of the *CYP24A1* gene was associated with the risk of developing HBP [43]. 

This study has several limitations, and a major limitation is the sample size since a larger cohort might make it possible to detect associations between the genetic variants being studied and the risk of HBP with greater statistical power. Moreover, a larger number of cases would enable us to strengthen the statistically significant causal relationship that was obtained. In addition, other limitations should be mentioned, such as the inherent ones from the retrospective studies, the ethnic differences and geographical variability in the studies with which the results are being compared, together with the lack of studies confirming an association between the *GC* rs7041 SNP and the risk of HBP.

The results shown in this study suggested that the presence of the T allele in the *GC* rs7041 SNP could be a protective factor against HBP. However, these results must be interpreted with caution, given the need for more scientific evidence to enable us to determine which of these SNPs could be used as biomarkers for the risk of HBP. Continued research with a larger cohort is still needed. The strengths of our study were the homogeneity of the sample, especially in terms of the cases, which consisted of only University Hospital Virgen de las Nieves patients diagnosed by the same team and also from the same geographical area, thus increasing uniformity. 

## 4. Materials and Methods

We conducted an observational retrospective case-control study.

### 4.1. Study Subjects

This study included 250 individuals diagnosed with HBP and 500 controls with no HBP diagnosis, such that both cases and controls were randomly enrolled, of Caucasian origin, from southern Spain. Subjects were followed up retrospectively from the enrolment. The inclusion criteria for the cases were age 18 years or over, diagnosis of HBP at the Hospital Universitario Virgen de las Nieves (between 1998 and 2018), and available clinical data. The controls were individuals aged over 18 years with no HBP diagnosis who had lived in the same geographical area and were recruited at the same hospital.

This case-control study was carried out in accordance with the Declaration of Helsinki, with the approval of the Ethics and Research Committee of the Andalusian Public Health Service’s Biobank (Code: 0957-N-21). The subjects signed a written informed consent form for the collection of blood or saliva samples and their donation to the Biobank. The samples were coded and treated confidentially.

### 4.2. Sociodemographic and Clinical Variables

The sociodemographic data compiled included sex, age at diagnosis, weight, height, and drinking status. From the weight and height values, the BMI was calculated and classified into three categories, according to the WHO scale (normal weight for BMI < 25 kg/m^2^, overweight for BMI ≥ 25 kg/m^2^ and <30 kg/m^2^, and obesity for BMI ≥ 30 kg/m^2^, respectively). Individuals were classified by standard drink units (SDUs) as non-drinkers if they were teetotalers or did not consume alcohol regularly, as current drinkers if their alcohol consumption was greater than 4 SDUs per day in men and greater than 2.5 SDUs per day in women, and as ex-drinkers, if their alcohol consumption had been greater than 4 SDUs per day in men and greater than 2.5 SDUs in women, but they did not currently drink [1]. The HBP diagnosis in the patients was performed in accordance with the criteria of the European Society of Cardiology and the European Society of Hypertension [1,2]. Cardiovascular morbidity was assessed through the presence of other pathologies, such as diabetes diagnosis (yes/no) and dyslipidemia diagnosis (yes/no), according to the criteria of the Clinical Practice Guidelines on the management of diabetes and dyslipidemia [44,45].

The clinical variables were collected at the time of diagnosis of the disease: SBP (normal < 120 mmHg; elevated 120–130 mmHg; high > 130 mmHg), DBP (normal < 80 mmHg; elevated 80–90 mmHg; high > 90 mmHg), and total cholesterol (normal < 200 mg/dL; elevated 200–240 mg/dL; high > 240 mg/dL), HDL cholesterol (at risk < 40 mg/dL; medium 40–60 mg/dL; optimum > 60 mg/dL), LDL cholesterol (normal < 130 mg/dL; elevated 130–160 mg/dL; high > 160 mg/dL), triglycerides (normal < 150 mg/dL; elevated 150–200 mg/dL; high > 200 mg/dL), and fasting blood sugar (mg/dL).

### 4.3. Genetic Variables

#### 4.3.1. DNA Isolation

Blood samples (3 mL) were collected in BD Vacutainer K3E Plus blood collection tubes, and saliva samples in BD Falcon 50 mL conical tubes (BD, Plymouth, UK). DNA was extracted using the QIAamp DNA Mini extraction Kit (Qiagen GmbH, Hilden, Germany), according to the manufacturer’s instructions for purification of DNA from blood or saliva, and stored at −40 °C. The concentration and purity of the DNA were measured using the NanoDrop 2000 UV spectrophotometer with 280/260 and 280/230 absorbance ratios. The DNA samples, isolated from blood or saliva, were preserved in the Biobank of the Hospital Universitario Virgen de las Nieves, part of the Andalusian Public Health Service’s Biobank.

#### 4.3.2. Detection of Gene Polymorphisms and Quality Control

The gene polymorphisms were determined by real-time polymerase chain reaction (PCR) allelic discrimination assay using TaqMan probes (ABI Applied Biosystems, QuantStudio 3 Real-Time PCR System, 96 wells), following the manufacturer’s instructions. Ten percent of the results were confirmed by Sanger sequencing (Table 5). Real-time PCR and Sanger sequencing were performed in the Pharmacogenetics Unit of the Hospital Universitario Virgen de las Nieves. The criteria for SNPs quality control were: (1) missing genotype rate per SNP < 0.05; (2) minor allele frequency > 0.01; (3) *p*-value > 0.05 in Hardy–Weinberg Equilibrium test; (4) missing genotype rate in the case group is less than 0.05, and in the control group is less than 0.05, respectively.

### 4.4. Statistical Analysis

Cases and controls were matched by age and sex using the 1:2 propensity score matching method [46]. Quantitative data were expressed as the result (± standard deviation) for variables with normal distribution and as medians or percentiles (25 and 75) for variables with non-normal distribution. The Shapiro–Wilks test was used to verify normality.

We determined the Hardy–Weinberg equilibrium and the haplotype frequency through the D’ and r^2^ coefficients. The bivariate association between risk of HBP and polymorphisms was evaluated for multiple models (genotypic, additive, allelic, dominant, and recessive), using the Pearson χ^2^ and Fisher’s exact tests, and assessed with the ORs and their corresponding 95% CIs. We defined the models as follows: genotypic (DD vs. Dd vs. dd), allelic (D vs. d), dominant (DD + Dd vs. dd), recessive (DD vs. Dd + dd), and additive (dd = 0, Dd = 1, DD = 2), where D is the minor allele and d the major allele. We considered unconditional multiple logistic regression models (genotypic, dominant, and recessive) to determine the influence of possible confounding variables on the risk of suffering from HBP.

All the tests were 2-sided, with a significance level of *p* < 0.05, and were estimated using PLINK and R 4.0.2 software [47,48]. We performed LD with Haploview 4.2 [49] and haplotype analysis with SNPStats [50].

## 5. Conclusions

This study shows that the *GC* rs7041 SNP is associated with hypertension development and could play a notable role as a biomarker for the risk of the disease. By contrast, no relationship was found between the rs1544410 (*VDR* BsmI), rs11568820 (*VDR* Cdx2), rs2228570 (*VDR* FokI), rs7975232 (*VDR* ApaI), rs731236 (*VDR* TaqI), rs10741657 (*CYP2R1*), rs4646536 (*CYP27B1*), rs3782130 (*CYP27B1*), rs10877012 (*CYP27B1*), rs703842 (*CYP27B1*), rs6068816 (*CYP24A1*), and rs4809957 (*CYP24A1*) SNPs and the risk of suffering from HBP.

## Figures and Tables

**Figure 1 ijms-24-05974-f001:**
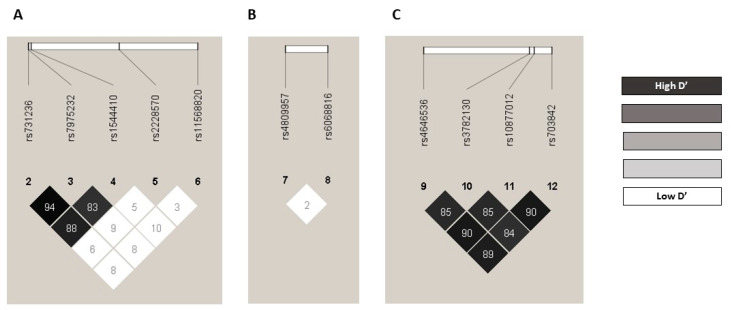
Pairwise LD D’ plots for SNPs located in *VDR* gene, *CYP24A1* gene, and *CYP27B1* gene in the whole population. Numbers inside the squares are the D’ values expressed as a percent: (**A**) pairwise LD D’ plots for 5 SNPs located in *VDR* gene, (**B**) pairwise LD D’ plots for 2 SNPs located in *CYP24A1* gene, and (**C**) pairwise LD D’ plots for 4 SNPs located in *CYP27B1* gene. LD: linkage disequilibrium; SNP: single nucleotide polymorphism.

**Table 1 ijms-24-05974-t001:** Clinicopathological characteristics of HBP cases and controls.

	Cases		Controls		χ²	*p*-Value	Reference	OR	95% CI
	N	n (%)	N	n (%)					
Sex	250		500						
Women		131 (52.4)		267 (53.4)	0.067	0.796			
Men		119 (47.6)		233 (46.6)					
Age	250	66 (60,73)	500	65 (60,73)		0.989 *			
Alcohol consumption	229		452						
Current drinkers		61 (26.6)		83 (18.4)	7.22	0.027	Non-drinkers	1.58	1.08–2.31
Ex-drinkers		5 (2.2)		18 (4.0)				0.60	0.19–1.53
Non-drinkers		163 (71.2)		351 (77.7)				1	
BMI	211		339						
Normal weight		34 (16.1)		111 (32.7)	24.86	<0.001	Normal weight	1	
Overweight		83 (39.3)		135 (39.8)				2.01	1.26–3.24

Obesity		94 (44.5)		93 (27.4)				3.30	2.06–5.38
Dyslipidemia	250		500						
Yes		123 (49.2)		133 (26.6)	37.86	<0.001	No	2.67	1.95–3.68
No		127 (50.8)		367 (73.4)				1	

Diabetes	250		487						
Yes		104 (41.6)		53 (10.9)	92.98	<0.001	No	5.83	4–8.58
No		146 (58.4)		434 (89.1)				1	
Systolic BP (mmHg)	250	137 (121,154)							
Normal		61 (24.4)							
Elevated		43 (17.2)							
High		146 (58.4)							
Diastolic BP (mmHg)	250	80 (70,88)							
Normal		146 (58.4)							
Elevated		58 (23.2)							
High		46 (18.4)							
Total Cholesterol (mg/dL)	249	195 ± 46							
Normal		137 (55)							
Elevated		76 (30.5)							
High		39 (14.5)							
HDL Cholesterol (mg/dL)	227	49 (41,60)							
Optimum		53 (23.3)							
Medium		121 (53.3)							
At risk		53 (23.3)							
LDL Cholesterol (mg/dL)	223	118 ± 37							
Normal		142 (63.7)							
Elevated		53 (23.8)							
High		28 (12.6)							
Triglycerides (mg/dL)	247	126 (94,166)							
Normal		171 (69.2)							
Elevated		45 (18.2)							
High		31 (12.6)							
Glucose (mg/dL)	250	98 (86,124)							

Qualitative variables: number (percentage). Quantitative variables: Normal distribution: mean ± standard deviation. Non-normal distribution: P_50_ [P_25_,P_75_]; * *p*-value for *t*-test. Shade means the value is significant. BP: blood pressure; HDL: high-density lipoprotein; LDL: low-density lipoprotein; OR: odds ratio; CI: confidence interval.

**Table 2 ijms-24-05974-t002:** Influence of *GC* rs7041 SNP on risk of HBP.

Model	Genotype	Cases (n (%))	Controls (n (%))	*p*-Value ^a^	OR	95% CI
Genotypic	TT	44 (18.2)	114 (23.5)	0.089	0.61	0.39–0.95
	TG	122 (50.4)	252 (51.9)		0.76	0.53–1.10
	GG	76 (31.4)	120 (24.7)		1	
Dominant	TT + TG	166 (68.6)	366 (75.3)	0.054	0.72	0.51–1.00
	GG	76 (31.4)	120 (24.7)			
Recessive	TT	44 (18.2)	114 (23.5)	0.104		
	GG + TG	198 (81.8)	372 (76.5)			
Allelic	T	210 (43.4)	480 (49.4)	0.031	0.79	0.63–0.98
	G	274 (56.6)	492 (50.6)			
Additive	-	-	-	0.028	0.78	0.62–0.97

^a^*p*-value for χ^2^ test. Shade means the value is significant. OR: odds ratio; CI: confidence interval; HBP: high blood pressure; SNP: single nucleotide polymorphism.

**Table 3 ijms-24-05974-t003:** Influence of clinical characteristics and *GC* rs7041 SNP on risk of HBP.

	Genotypic						Dominant			Recessive			Additive		
	TT vs. GG			TG vs. GG			TT + TG vs. GG			TT vs. TG + GG			GG = 0, TG = 1, TT = 2		
	*p*-Value	OR	95% CI	*p*-Value	OR	95% CI	*p*-Value	OR	95% CI	*p*-Value	OR	95% CI	*p*-Value	OR	95% CI
Overweight	0.018	1.84	1.12–3.07	0.018	1.84	1.12–3.07	0.018	1.84	1.12–3.06	0.019	1.82	1.11–3.03	0.018	1.84	1.12–3.07
Obesity	0.001	2.40	1.43–4.08	0.001	2.40	1.43–4.08	<0.001	2.42	1.44–4.11	0.002	2.29	1.37–3.87	0.001	2.40	1.43–4.08
Diabetes	<0.001	3.54	2.25–5.65	<0.001	3.54	2.25–5.65	<0.001	3.44	2.19–5.46	<0.001	3.63	2.31–5.79	<0.001	3.52	2.24–5.61
Dyslipidemia	0.002	1.86	1.26–2.77	0.002	1.86	1.26–2.77	0.002	1.84	1.24–2.74	0.002	1.83	1.24–2.71	0.002	1.87	1.27–2.78
rs7041	0.005	0.44	0.41–0.77	0.045	0.64	0.41–0.99	0.010	0.69	0.47–1.03	0.035	0.58	0.35–0.95	0.003	0.65	0.4–0.87

Shade means the value is significant. OR: odds ratio; CI: confidence interval; HBP: high blood pressure; SNP: single nucleotide polymorphism.

**Table 4 ijms-24-05974-t004:** Influence of haplotypes formed by 6 SNPs located on Chromosome 12 on risk of HBP.

ID	rs1544410 (*VDR*)	rs7975232 (*VDR*)	rs731236 (*VDR*)	rs4646536 (*CYP27B1*)	rs703842 (*CYP27B1*)	rs10877012 (*CYP27B1*)	Freq.	OR (95% CI)	*p*-Value
1	G	C	T	A	T	G	0.3222	1.00	---
2	A	A	C	A	T	G	0.2356	0.86 (0.60–1.23)	0.41
3	A	A	C	G	C	T	0.1173	1.07 (0.67–1.70)	0.79
4	G	A	T	A	T	G	0.0973	0.83 (0.51–1.35)	0.46
5	G	C	T	G	C	T	0.0745	1.00 (0.55–1.81)	1
6	A	C	T	A	T	G	0.0211	1.45 (0.50–4.20)	0.5
7	G	A	T	G	C	T	0.0162	1.25 (0.30–5.14)	0.76
8	G	A	C	A	T	G	0.0159	0.35 (0.12–1.02)	0.054
9	A	A	T	A	T	G	0.0137	1.00 (0.31–3.25)	1
Rare	*	*	*	*	*	*	0.0862	1.44 (0.82–2.53)	0.2

Freq.: haplotype frequency in the whole population. * Refer to those rare haplotypes with haplotype frequency < 0.01 in the whole population, as there is no symbol to identify this group. Shade means the value is significant. OR: odds ratio; CI: confidence interval; HBP: high blood pressure; SNP: single nucleotide polymorphism.

**Table 5 ijms-24-05974-t005:** Gene SNPs and their TaqMan Assay IDs.

Gene	Location, SNP	dbSNP ID	Assay ID
*VDR* (12q13.11)	Intron 8, G > A	rs1544410 (BsmI)	AN324M4 *
	Intron 1, G > A	rs11568820 (Cdx2)	C___2880808_10
	Exon 2, C > T	rs2228570 (FokI)	C__12060045_20
	Intron 8, C > A	rs7975232 (ApaI)	C__28977635_10
	Exon 9, T > C	rs731236 (TaqI)	C___2404008_10
*CYP27B1* (12q14.1)	Intron 6, T > C	rs4646536	C__25623453_10
	Promoter 5’, G > A/G > C	rs3782130	ANGZRHH *
	5’ UTR, A > G 3’ UTR, A > G	rs10877012 rs703842	C__26237740_10 ANH6J3F *
*CYP24A1* (20q13.2)	Exon 6, G > A	rs6068816	C__25620091_20
	3’ UTR, A > G	rs4809957	C___3120981_20
*GC* (4q13.3)	Exon 11, T > G	rs7041	C___3133594_30
*CYP2R1* (11p15.2)	5’ UTR, A > G	rs10741657	C___2958430_10

* The SNPs were analyzed using custom assays by ThermoFisher Scientific (Waltham, MA, USA). SNP: single nucleotide polymorphism; UTR, untranslated region.

## Data Availability

Data unavailable due to privacy and ethical restrictions.

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
