# Peer review of "Single Nucleotide Polymorphisms in the Vitamin D Metabolic Pathway and Their Relationship with High Blood Pressure Risk"

_ijms, 2023, doi:10.3390/ijms24065974_

Round 1
Reviewer 1 Report
The authors revised carefully the manuscript, and should be published in the present form.
Reviewer 2 Report
Authors have investigated the relationship between single nucleotide polymorphisms in the Vitamin D Metabolic
Pathway and high blood pressure. They found that a SNP within the GC gene significantly modulated the risk for high blood pressure.
Although results are intriguing, the study presents a series of flaws that need to be corrected.
- It is not clear to this reviewer how cases and controls were enrolled. Authors should report the time span for the enrollment of different subsettings. In addition, because of the huge differences between cases and controls, which criteria were applied to choose controls? Had cases or controls cardiovascular diseases?
- Whether the GC SNP affects the risk for arterial hypertension, it would be interesting to check for an association with blood pressure values (systolic, diastolic, mean) in the control setting, i.e. by means of a multiple regression analysis.
- Because a high number of associations was investigated, it would be meaningful to apply some statistics to take into account the risk of a false positive results (i.e. Bonferroni test).
- Fig.1 could be misleading for readers, because the draft in the upper panel indicates a chromosome and seems to indicate that a linkage was investigated between SNPs in distinct genes located in different chromosomes.
Authors have investigated the relationship between single nucleotide polymorphisms in the Vitamin D Metabolic
Pathway and high blood pressure. They found that a SNP within the GC gene significantly modulated the risk for high blood pressure.
Although results are intriguing, the study presents a series of flaws that need to be corrected.
- It is not clear to this reviewer how cases and controls were enrolled. Authors should report the time span for the enrollment of different subsettings. In addition, because of the huge differences between cases and controls, which criteria were applied to choose controls? Had cases or controls cardiovascular diseases?
- Whether the GC SNP affects the risk for arterial hypertension, it would be interesting to check for an association with blood pressure values (systolic, diastolic, mean) in the control setting, i.e. by means of a multiple regression analysis.
- Because a high number of associations was investigated, it would be meaningful to apply some statistics to take into account the risk of a false positive results (i.e. Bonferroni test).
- Fig.1 could be misleading for readers, because the draft in the upper panel indicates a chromosome and seems to indicate that a linkage was investigated between SNPs in distinct genes located in different chromosomes.
Authors have investigated the relationship between single nucleotide polymorphisms in the Vitamin D Metabolic
Pathway and high blood pressure. They found that a SNP within the GC gene significantly modulated the risk for high blood pressure.
Although results are intriguing, the study presents a series of flaws that need to be corrected.
- It is not clear to this reviewer how cases and controls were enrolled. Authors should report the time span for the enrollment of different subsettings. In addition, because of the huge differences between cases and controls, which criteria were applied to choose controls? Had cases or controls cardiovascular diseases?
- Whether the GC SNP affects the risk for arterial hypertension, it would be interesting to check for an association with blood pressure values (systolic, diastolic, mean) in the control setting, i.e. by means of a multiple regression analysis.
- Because a high number of associations was investigated, it would be meaningful to apply some statistics to take into account the risk of a false positive results (i.e. Bonferroni test).
- Fig.1 could be misleading for readers, because the draft in the upper panel indicates a chromosome and seems to indicate that a linkage was investigated between SNPs in distinct genes located in different chromosomes.
Authors have investigated the relationship between single nucleotide polymorphisms in the Vitamin D Metabolic
Pathway and high blood pressure. They found that a SNP within the GC gene significantly modulated the risk for high blood pressure.
Although results are intriguing, the study presents a series of flaws that need to be corrected.
- It is not clear to this reviewer how cases and controls were enrolled. Authors should report the time span for the enrollment of different subsettings. In addition, because of the huge differences between cases and controls, which criteria were applied to choose controls? Had cases or controls cardiovascular diseases?
- Whether the GC SNP affects the risk for arterial hypertension, it would be interesting to check for an association with blood pressure values (systolic, diastolic, mean) in the control setting, i.e. by means of a multiple regression analysis.
- Because a high number of associations was investigated, it would be meaningful to apply some statistics to take into account the risk of a false positive results (i.e. Bonferroni test).
- Fig.1 could be misleading for readers, because the draft in the upper panel indicates a chromosome and seems to indicate that a linkage was investigated between SNPs in distinct genes located in different chromosomes.
Author Response
Comments to the reviewers can be found in the attached file.

Reviewer 3 Report
The authors examined the associations between SNPs in genes which encode proteins involved in vitamin D metabolism and high blood pressure.
The objective is important because high blood pressure represents one of the most significant causes for cardiovascular diseases and vitamin D deficiency is very frequent state in the developed countries and accordingly a likely important contributor to hypertension.
The study is is well designed and performed.
The authors found association between the genetic polymorphism in the GC but not other tested genes.
Obviously it was not the aim of the study, but having vit D3 levels of the study groups would further increase the significance of the study.
Issues:
Table 1: There are no data for the controls on SBP, DBP, total cholesterol, HDL-C, LDL-C, TG, Glucose. There is no explanation for the missing data.
How without these data the authors know the frequency of diabets and dyslipidemina in the control group.
Lane 124: ''for'' sex instead ''in'' sex; there are several similar mistakes throughout the text.
Author Response

(The authors gave the same response as above.)

Reviewer 4 Report
Proposed paper is interesting and well written. However, some revisions are needed before it can be accepted for publication:
- Continue variables in table 1 (SBP, DBP, cholesterol and so on) should be expressed as mean and standard deviation and not categorized in normal, elevated and high.
- Since this is a cross sectional study authors cannot state that "polymorphism located in the GC gene was significantly associated with a lower risk of developing 37 HBP" but that it is associated with a lower prevalence of HBP. This is a really important difference. On the same line the latest sentence of the ABS is not true. Also the discussion should be changed accordingly.
- What about therapies for hypertension? this could probaly have an effect on vitamin D(diuretics in particular) that should be discussed in the relative section and they need to be added to the multivariate model and stated in table 1.
- What the polimorphism founded to be associated with BP does on vitamin D? it increase the level or it decrease it? or even it is not known? this should be clearly stated in the discussion but also in the abstract.
- An interesting review on the argument has been forgottenand should be cited (10.1007/s40292-014-0060-5)
Author Response

(The authors gave the same response as above.)

Round 2
Reviewer 2 Report
Authors have revised their manuscript and it is now ameliorated.
I have some difficulties with the last raw of table 3, the shaded one. I read some inconsistencies between OR, CIs and the corresponding significant value: how an OR of 0.65 (95%-CI:1.00–1.75) can display a so high significant level (0.003)? It can be anticipated that for CIs laying below and upper the unit, the significance should be >0.05.
Reviewer 3 Report
The authors significantly improved the manuscript.
Reviewer 4 Report
Authors replies to all the query raised and paper improves and can now be accepter for publication
